# Improving Vertical Federated Learning by Efficient Communication with ADMM

## Abstract

Vertical Federated learning (VFL) allows each client to collect partial features and jointly train the shared model. In this paper, we identified two challenges in VFL: (1) some works directly average the learned feature embeddings and therefore might lose the unique properties of each local feature set; (2) server needs to communicate gradients with the clients for *each* training step, incurring high communication cost. We aim to address the above challenges and propose an efficient VFL with multiple heads (`VIM`) framework, where each head corresponds to local clients by taking the separate contribution of each client into account. In addition, we propose an Alternating Direction Method of Multipliers (ADMM)-based method to solve our optimization problem, which reduces the communication cost by allowing multiple local updates in each step. We show that `VIM` achieves significantly higher accuracy and faster convergence compared with state-of-the-arts on four datasets, and the weights of learned heads reflect the importance of local clients.

## 1 Introduction

Federated learning (FL) has enabled large-scale training with data privacy guarantees on distributed data for different applications [34, 3, 12, 33, 31]. In general, FL can be categorized into Horizontal FL (HFL) [24] where data samples are distributed across clients, and Vertical FL (VFL) [31] where features of the samples are partitioned across clients and the labels are usually owned by the server (or the active party in two-party setting [13]). In particular, VFL allows agents with partial information of the same dataset to jointly train the model, which leads to many real-world applications [16, 31, 12].

Despite the importance and practicality of VFL, there are mainly two weaknesses of the state-of-the-art (SOTA) VFL frameworks: (1) some VFL frameworks directly average the feature embeddings from local agents, and therefore fail to capture the unique properties of each local feature set [4]; (2) the server usually needs to send gradients to clients for *each* training step which leads to high communication cost and potentially rapid consumption of privacy budget [4, 16, 19].

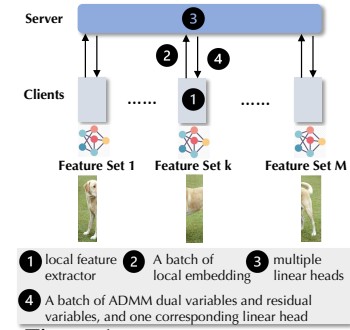

Figure 1: Overview of `VIMADMM`

To solve the above challenges, in this work, we propose an efficient VFL optimization framework with multiple heads (`VIM`), where each head corresponds to one local client, taking the individual contribution of clients into consideration and thereby improving the overall performance. In particular, we propose an Alternating Direction Method of Multipliers (ADMM) [2]-based method to solve our optimization problem, which allows multiple local updates in each step, thus yielding faster convergences and reducing the communication cost. This is critical to preserving privacy since the privacy costs increases as the number of communication rounds increases [1]. We consider various VFL settings including *with model splitting* (i.e., clients host partial models) and *without model splitting* (i.e., clients hold the entire model). Under the with model

Submitted to 36th Conference on Neural Information Processing Systems (NeurIPS 2022). Do not distribute.

splitting setting, we propose the gradient-based algorithm `VIMSGD` as well as the ADMM-based algorithm `VIMADMM` under `VIM` framework. Compared to gradient-based methods, `VIMADMM` not only reduces communication frequency but also reduces the dimensionality by only exchanging ADMM-related variables. With modifications of communication strategies and updating rules for servers and clients, we extend `VIMADMM` to the without model splitting setting and propose `VIMADMM-J`. Moreover, we show that a byproduct of `VIM` is that the weights of learned linear heads reflect the importance of local clients, which enables functionalities such as client-level explanation, client denoising and client summarization. Our technical contributions are:

- We propose an efficient and effective VFL optimization framework with multiple heads (`VIM`). To solve our optimization problem, we propose an ADMM-based method, which reduces communication costs by allowing multiple local updates at each step.
- We conduct extensive experiments on MNIST, CIFAR, NUS-WIDE, and ModelNet40 datasets, and show that ADMM-based algorithms under `VIM` converge faster, achieve higher accuracy than existing VFL frameworks.
- We evaluate our client-level explanation under `VIM` based on the linear heads weights norm, and demonstrate the functionalities it enables such as clients denoising and summarization.

## 2 Related Work

**Vertical Federated Learning.** VFL has been well studied for simple models including trees [5, 30], kernel models [11], and linear and logistic regression [13, 32, 36, 9, 15, 23]. For DNNs, there are two popular VFL settings: with model splitting [29, 19, 4] and without model splitting [16, 18]. In the model splitting setting, split learning [29] is the first related paradigm, where each client trains a partial network up to a cut layer, the server concatenates local activations and trains the rest of the network. However, despite its promising performance in HFL, it was not evaluated on vertically partitioned data. VAFL [4] is proposed for VFL where the server averages the local embeddings and sends gradients back to clients to update local models. However, such embedding averaging might lose the unique properties of each client. FedMVT [19] focuses on the semi-supervised VFL with multi-view learning. For VFL without model splitting setting, in FDML [16] framework, each client submits local logits to the server, who averages over the logits and send gradients back to clients. We note that all SOTA methods [4, 19, 16] require the communication of gradients at *each* training step, leading to high communication costs before convergence.

## 3 VFL with Multiple Heads (`VIM`)

### 3.1 Framework Overview

In VFL, we have $M$ clients $\{1, 2, \ldots, M\}$ who hold different feature sets of the same training samples to jointly train a machine learning model. We consider the classification task and denote $d_c$ as the number of classes. Suppose there is a training dataset $D = \{x_j, y_j\}_{j=1}^N$ containing $N$ samples, the server owns the labels $\{y_j\}_{j=1}^N$, and each client $k$ has a local feature set $X_k = \{x_j^k\}_{j=1}^N$, where the vector $x_j^k \in \mathbb{R}^{d^k}$ denotes the local (partial) features of sample $j$. The overall feature $x_j \in \mathbb{R}^d$ of sample $j$ is the concatenation of all local features $\{x_j^1, x_j^2, \ldots, x_j^M\}$, with $d = \sum_{k=1}^M d^k$.

Due to the privacy protection requirement of VFL, each client $k$ does not share raw local feature set $X_k$ with other clients or the server. Instead, VFL consists of two steps: (1) *local processing step*: each client learns a local model that maps the local features to local outputs and sends them to the server. (2) *server aggregation step*: the server aggregates the local outputs from all clients to compute the final prediction for each sample as well as the corresponding losses. Depending on whether or not the server holds a model, there are two popular VFL settings [10]: VFL *with model splitting* [4, 29] and VFL *without model splitting* [16]: (i) In the model splitting setting, each client trains a feature extractor as the local model that outputs *local embeddings*, and the server owns a model which predicts the final results based on the aggregated embeddings. (ii) In the VFL without model splitting setting, the clients host the entire model that outputs the *local logits*, and the server simply performs the logits aggregation operation without hosting any model. In both settings, the local model is updated by federated backward propagation [10]: a) the server first computes the gradients of the loss w.r.t the local output (either embeddings or logits) from each client separately and sends the gradients back to clients; b) each client calculates the gradients of local output w.r.t the local model parameters and updates the local model using the chain rule.

We will first dive into the details of the model splitting setting and introduce our framework `VIM` as well as the corresponding SGD-based method `VIMSGD` and ADMM-based method `VIMADMM`. Then,

we will show that our ADMM-based method can be easily extended to the VFL without model splitting setting with slight modifications, which is based on different communication strategies and update rules for server and clients, yielding the method `VIMADMM-J`.

## 3.2 VFL with Model Splitting

**Setup.** Let $f$ parameterized by $\theta_k$ be the local model (i.e., feature extractor) of client $k$, which outputs a local embedding vector $h_j^k = f(x_j^k; \theta_k) \in \mathbb{R}^{d_f}$ for each local feature $x_j^k$. We denote the parameters of the model on the server-side as $\theta_0$. Overall, the clients and the server aim to collaboratively solve the Empirical Risk Minimization (ERM) objective:

$$\min_{\{\theta_k\}_{k=1}^M, \theta_0} \sum_{j=1}^N \ell(\{h_j^1, \ldots, h_j^M\}, y_j; \theta_0) + \sum_{k=1}^M \beta \mathcal{R}(\theta_k) + \beta \mathcal{R}(\theta_0) \quad \text{with } h_j^k = f(x_j^k; \theta_k), \forall k \in [M], \quad (1)$$

where $\ell$ is a loss function (e.g., cross-entropy loss with softmax function), $\mathcal{R}$ is a regularizer on model parameters, and $\beta \in \mathbb{R}$ is the regularization weight for client $k$ or the server. We consider $\mathcal{R}$ to be differentiable but are optimistic that it can be extended to other regularizers as future work. In principle, $\beta$ can be different for different models, and we use the same $\beta$ here for simplicity.

**VIM Formulation.** We start by noting that for the server aggregation step, the SOTA method, VAFL [4], directly averages the local embeddings $\sum_{k=1}^M \alpha_k h_j^k$ where the scalar $\alpha_k \in \mathbb{R}$ is the aggregation weight for client $k$ and it can be optimized during training as an additional parameter in the ERM loss. Therefore, the objective function of VAFL is $\min_{\{\theta_k\}_{k=1}^M, \{\alpha_k\}_{k=1}^M, \theta_0} \sum_{j=1}^N \ell(\sum_{k=1}^M \alpha_k h_j^k, y_j; \theta_0) + \sum_{k=1}^M \beta \mathcal{R}(\theta_k) + \beta \mathcal{R}(\theta_0)$. However, such an aggregation implicitly assumes that each dimension of the embedding vectors from different clients shares the same contextual meaning in the latent space so that they can be directly averaged. Such a design may be suboptimal, since in VFL different local embeddings can represent different aspects of the same sample, and therefore average-based aggregation might lose the unique properties of each local feature set.

To address the above average-based aggregation problem, we propose `VIM`, a novel VFL framework where the server learns a model with multiple linear heads corresponding to local clients, taking the separate contribution of each client into account. Specifically, the server's model $\theta_0$ consists of $M$ linear heads $W_1, W_2, \ldots, W_M$ with $W_k \in \mathbb{R}^{d_f \times d_c}, k \in [M]$, and the server's model outputs $\sum_{k=1}^K h_j^k W_k$ as the prediction for sample $j$, yielding our `VIM` objective:

$$\min_{\{W_k\}_{k=1}^M, \{\theta_k\}_{k=1}^M} \mathcal{L}_{\text{VIM}}(\{W_k\}, \{\theta_k\}) := \sum_{j=1}^N \ell(\sum_{k=1}^M f(x_j^k; \theta_k) W_k, y_j) + \sum_{k=1}^M \beta_k \mathcal{R}_k(\theta_k) + \sum_{k=1}^M \beta_k \mathcal{R}_k(W_k) \quad (2)$$

Despite the simplicity of linear heads, recent studies in representation learning show that the linear classifier is an efficient approach to predicting the labels on top of embedding representations [26, 20], given the expressive power of the local feature extractor which captures essential information from raw feature sets.

**VIMSGD.** Existing VFL frameworks often use SGD to alternatively update the server's model and local models [4, 19] where the clients send a *batch* of embeddings to the server, and the server sends a *batch* of gradients to clients at each communication round. We provide the SGD-based algorithm `VIMSGD` under the `VIM` framework (The Algorithm 1 and detailed description are deferred to Appendix A), which serves as a strong baseline.

**VIMADMM.** The SGD-based methods including the state-of-art VAFL require the server to send the *gradients* w.r.t embeddings back to clients at *every* training step of the local models. However, such (1) frequent communication and (2) the high dimensionality of gradients (i.e., $bd_f$ for $b$ samples) lead to high communication costs. To address the above limitations, we propose an ADMM-based method for `VIM`, reducing the communication frequency by allowing multiple local updates at each round, and reducing the dimensionality by only exchanging ADMM-related variables (i.e., $(2b + d_f)d_c$ for $b$ samples where $d_c \ll d_f, b$ for most VFL settings today [4, 16]). Specifically, we note that Eq. 2 can be viewed as the *sharing problem* (e.g., [2, Section 7.3]) involving each agent adjusting its variable to minimize its individual cost $\mathcal{R}(\theta_k) + \mathcal{R}(W_k)$, as well as the shared objective term $\ell(\sum_{k=1}^M h_j^k W_k, y_j)$. Moreover, the multiple heads in `VIM` enable the application of ADMM via a special decomposition into simpler sub-problems that can be solved in a distributed manner. We begin by rewriting Eq. 2 to an equivalent constrained optimization problem by introducing auxiliary variables $z_1, z_2, \ldots, z_N \in \mathbb{R}^{d_c}$:

$$\min_{\{W_k\}_{k=1}^M, \{\theta_k\}_{k=1}^M, \{z_j\}_{j=1}^N} \sum_{j=1}^N \ell(z_j, y_j) + \sum_{k=1}^M \beta_k \mathcal{R}_k(\theta_k) + \sum_{k=1}^M \beta_k \mathcal{R}_k(W_k) \text{ s.t. } \sum_{k=1}^M f(x_j^k; \theta_k) W_k - z_j = 0, \forall j \in [N]. \quad (3)$$

Notably, each linear constraint implies a consensus between the server's output $\sum_{k=1}^{M} h_j^k W_k$ and the auxiliary variable $z_j$ for each sample $j$. The augmented Lagrangian which adds a quadratic term to the Lagrangian of Eq. 3 is given by:

$$\min_{\{W_k\}_{k=1}^M, \{\theta_k\}_{k=1}^M, \{z_j\}_{j=1}^N, \{\lambda_j\}_{j=1}^N} \mathcal{L}_{\text{ADMM}}(\{W_k\}_{k=1}^M, \{\theta_k\}_{k=1}^M, \{z_j\}_{j=1}^N, \{\lambda_j\}_{j=1}^N) := \sum_{j=1}^N \ell(z_j, y_j) \quad (4)$$

$$+ \sum_{k=1}^{M} \beta_k \left( \mathcal{R}_k(\theta_k) + \mathcal{R}_k(W_k) \right) + \sum_{j=1}^{N} \lambda_j^\top \left( \sum_{k=1}^{M} f(x_j^k; \theta_k) W_k - z_j \right) + \frac{\rho}{2} \sum_{j=1}^{N} \left\| \sum_{k=1}^{M} f(x_j^k; \theta_k) W_k - z_j \right\|_F^2,$$

where $\lambda_j \in \mathbb{R}^{d_c}$ is the dual variable for sample $j$, and $\rho \in \mathbb{R}^+$ is a constant penalty factor. To solve Eq. 4, we follow the standard ADMM algorithm [2] and update the primal variables $\{W_k\}$, $\{\theta_k\}$, $\{z_j\}$ and the dual variables $\{\lambda_j\}$ *alternatively* as in Eq. 5, which decomposes the problem in Eq. 3 into four sets of sub-problems over $\{W_k\}$, $\{\theta_k\}$, $\{z_j\}$, $\{\lambda_j\}$, and each sub-problem can be solved *in parallel*. In practice, we propose the following strategy for the alternative updating in the server and clients: (i) updating $\{z_j\}$, $\{\lambda_j\}$ and $\{W_k\}$ at server-side, (ii) updating $\{\theta_k\}$ at the client-side in parallel. Moreover, we

$$W_k^{(t+1)} = \underset{W_k}{\arg\min} \mathcal{L}(\{\theta_{k'}^{(t)}\}, W_k, \{z_j^{(t)}\}, \{\lambda_j^{(t)}\}), \forall k \in [M],$$

$$\theta_k^{(t+1)} = \underset{\theta_k}{\arg\min} \mathcal{L}(\theta_k, \{W_{k'}^{(t+1)}\}, \{z_j^{(t)}\}, \{\lambda_j^{(t)}\}), \forall k \in [M],$$

$$z_j^{(t+1)} = \underset{z_j}{\arg\min} \mathcal{L}(\{\theta_k^{(t+1)}\}, \{W_k^{(t+1)}\}, z_j, \{\lambda_{j'}^{(t)}\}), \forall j \in [N],$$

$$\lambda_j^{(t+1)} = \underset{\lambda_j}{\arg\min} \mathcal{L}(\{\theta_k^{(t+1)}\}, \{W_k^{(t+1)}\}, \{z_{j'}^{(t+1)}\}, \lambda_j), \forall j \in [N],$$

$$(5)$$

consider the realistic setting of stochastic ADMM with mini-batches. Concretely, at communication round $t$, the server samples a set of data indices, $B(t)$, with batch size $|B(t)| = b$. Then we describe the key steps of VIMADMM as follows:

(1) *Communication from client to server.* Each client $k$ sends a batch of embeddings $\{h_j^{k(t)}\}_{j \in B(t)}$ to the server, where $h_j^{k(t)} = f(x_j^k; \theta_k^{(t)})$, $\forall j \in B(t)$.

(2) *Sever updates auxiliary variables* $\{z_j\}$. After receiving the local embeddings from all clients, the server updates the auxiliary variable for each sample $j$ as:

$$z_j^{(t)} = \underset{z_j}{\arg\min} \quad \ell(z_j, y_j) - \lambda_j^{(t-1)^\top} z_j + \frac{\rho}{2} \left\| \sum_{k=1}^{M} h_j^{k(t)} W_k^{(t)} - z_j \right\|_F^2, \forall j \in B(t) \quad (6)$$

Since the optimization problem in Eq. 6 is convex and differentiable with respect to $z_j$, we use the L-BFGS-B algorithm [37] to solve the minimization problem.

(3) *Sever updates dual variables* $\{\lambda_j\}$. The server updates the dual variable for each sample $j$ as:

$$\lambda_j^{(t)} = \lambda_j^{(t-1)} + \rho \left( \sum_{k=1}^{M} h_j^{k(t)} W_k^{(t)} - z_j^{(t)} \right), \forall j \in B(t) \quad (7)$$

(4) *Sever updates linear heads* $\{W_k\}$. Each linear head of the server is then updated as:

$$W_k^{(t+1)} = \underset{W_k}{\arg\min} \quad \beta \mathcal{R}(W_k) + \sum_{j \in B(t)} \lambda_j^{(t)^\top} h_j^{k(t)} W_k + \sum_{j \in B(t)} \frac{\rho}{2} \left\| \sum_{i \in [M], i \neq k} h_j^{i(t)} W_i^{(t)} + h_j^{k(t)} W_k - z_j^{(t)} \right\|_F^2, \forall k \in [M] \quad (8)$$

For squared $\ell_2$ regularizer $\mathcal{R}$, we solve $W_k^{(t+1)}$ in an inexact way to save the computation by *one* step of SGD with the objective of Eq. 8.

(5) *Communication from server to client.* After the updates in Eq. 8, we define a residual variable $s_j^{k(t+1)}$ for each sample $j$ of $k$-th client, which provides supervision for updating local model:

$$s_j^{k(t+1)} \triangleq z_j^{(t)} - \sum_{i \in [M], i \neq k} h_j^{i(t)} W_i^{(t+1)}, \forall j \in B(t), \forall k \in [M] \quad (9)$$

The server sends the dual variables $\{\lambda_j^{(t+1)}\}_{j \in B(t)}$ and the residual variables $\{s_j^{k(t+1)}\}_{j \in B(t)}$ of all samples, as well as the *corresponding* linear head $W_k^{(t+1)}$ to each client $k$.

(6) *Client updates local model parameters* $\theta_k$. Finally, every client $k$ locally updates the model parameters $\theta_k$ as follows:

$$\theta_k^{(t+1)} = \underset{\theta_k}{\arg\min} \quad \beta \mathcal{R}(\theta_k) + \sum_{j \in B(t)} \lambda_j^{(t+1)^\top} f(x_j^k; \theta_k) W_k^{(t+1)} + \frac{\rho}{2} \sum_{j \in B(t)} \left\| s_j^{k(t+1)} - f(x_j^k; \theta_k) W_k^{(t+1)} \right\|_F^2. \quad (10)$$

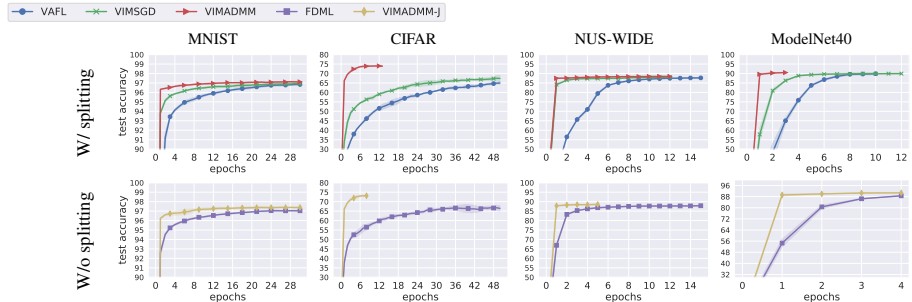

Figure 2: Performance comparison under w/ and w/o model splitting. Our methods outperform baselines.

Due to the nonconvexity of the loss function of DNN, we use $\tau$ local steps of SGD to update the local model at each round with the objective of Eq. 10. We note that multiple local updates of Eq. 10 enabled by ADMM lead to better local models at each communication round compared to gradient-based methods, thus VIMADMM requires fewer communication rounds to converge as we will show in Sec. 4.1. These six steps of VIMADMM are summarized in Algorithm 2 in Appendix A.

Note that ADMM auxiliary variables $\{z_j\}$ and dual variables $\{\lambda_j\}$ are only used during the training time optimization process. Therefore, in the test phase, for any sample $x_{j'}$, the server directly uses the trained multiple linear heads to make prediction $\sum_{k=1}^M h_{j'}^k W_k$.

### 3.3 VFL without Model Splitting

**Setup.** Recall the VFL without model splitting setting described in Section 3.1. Let $p$ parameterized by $\widetilde{\theta}_k$ be the local model (i.e., whole model) of client $k$, which outputs local logits $o_j^k = p(x_j^k; \widetilde{\theta}_k) \in \mathbb{R}^{d_c}$ for each local feature $x_j^k$. The clients and the server aim to jointly solve the problem:
$$\min_{\{\widetilde{\theta}_k\}_{k=1}^M} \sum_{j=1}^N \ell(\{o_j^1, \ldots, o_j^M\}, y_j) + \beta \sum_{k=1}^M \mathcal{R}(\widetilde{\theta}_k) \quad \text{with } o_j^k = p(x_j^k; \widetilde{\theta}_k), \forall k \in [M].$$

**VIMADMM-J.** In the state-of-art VFL framework FDML, the server averages the local logits as final prediction $\sum_{k=i}^M o_j^k$, and FDML also suffers from the high communication cost by sending the gradients w.r.t. local logits to each client at *each* training step of the local model. To solve this problem with our VIM framework, we adapt VIMADMM to the without model splitting setting and propose VIMADMM-J, where each linear head $W_k$ is held by the corresponding client $k$, and is always updated locally. The detailed description of key steps of VIMADMM-J and the corresponding Algorithm 3 are presented in Appendix A.

## 4 Experiments

In this section, we show that our proposed framework VIM achieves significantly faster convergence and higher accuracy than SOTA and enables client-level explainability on four real-world datasets.

**Data and Models.** We consider the classification task on four datasets: MNIST [22], CIFAR [21], NUS-WIDE [6], a multi-modality dataset with image features and textual features, and ModelNet40 [27], a multi-view image dataset. As shown in Figure 3 row 1, we simulate VFL scenarios by splitting the data features to $\{14, 9, 4, 4\}$ clients for the four datasets respectively. As for the local model, we use a two-layer fully connected model for MNIST and NUS-WIDE, a CNN model for CIFAR, and ResNet-18 [14] for ModelNet40. To prevent over-fitting, we adopt standard stopping criteria, i.e., stop training when the model converges or the validation accuracy starts to drop more than $2\%$. We refer to Appendix B for more details about datasets, networks, and parameter selection.

**Baselines.** We compare VIMSGD, VIMADMM with VAFL [4] under *w/ model splitting* , and compare VIMADMM-J with FDML [16] under *w/o model splitting* . Experiments are run 3 times.

### 4.1 Performance Evaluation under VFL

We observe from Figure 2 that three VIM algorithms consistently outperform baselines under VFL. Specifically, (1) VIMADMM and VIMSGD converge significantly faster than VAFL on four datasets and achieve higher accuracy than VAFL especially on CIFAR, which shows that the aggregation in VIM is better than embedding averaging as in VAFL by learning separate linear weights for each client. (2) ADMM-based methods converge faster than gradient-based methods. For example, on CIFAR, VIMADMM and VIMADMM-J achieves 73.85%, 73.12% at epoch 8, while VAFL, VIMSGD, FDML, only achieves 46.16%, 56.26%, 56.50% at epoch 50. This is because the multiple local updates enabled by ADMM lead to better local models at each round, thereby speeding up the convergence and reducing the communication costs. For instance, each epoch consists of 44 communication rounds on

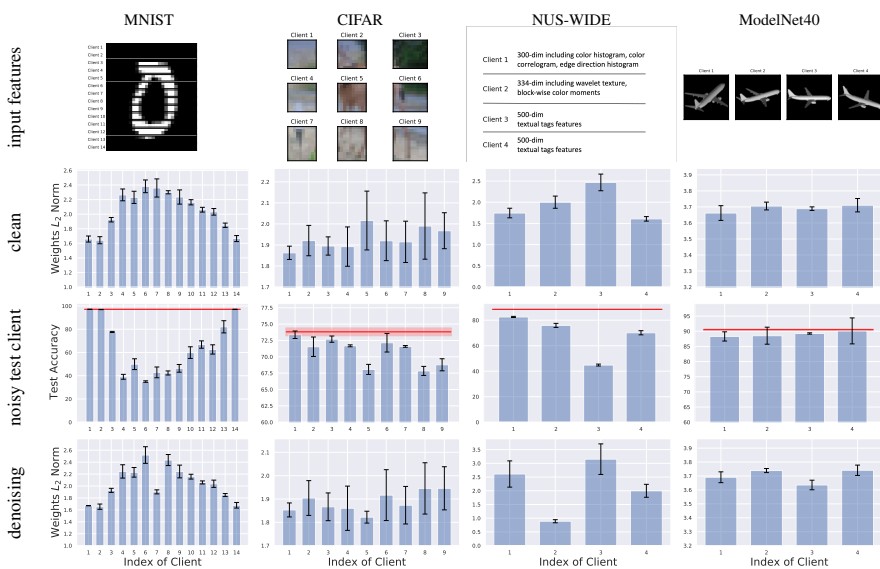

Figure 3: Input features for each client (row 1), the weights norm of linear heads under clean setting (row 2) and under one noisy client (row 4), and test accuracy when each client's test input features are perturbed (row 3) where red line denotes the test accuracy without perturbation.

CIFAR. Our `VIMADMM` takes 42 fewer epochs than VAFL to converge, which *saves more than 1.8k communication rounds in practice.* We defer the analysis of the effect of penalty factor $\rho$ and local steps $\tau$ on VIMADMM, and the communication cost comparison to Appendix B.

### 4.2 Client-level Explainability of `VIM`

We show that the weights of learned linear heads reflect the importance of local clients based on the weights norm histogram, which enables functionalities such as test-time noise validation and client denoising. We defer the results on client summarization and the visualization of the local embedding that justifies the design of `VIM` to the Appendix B.

**Client Importance.** Given a trained `VIMADMM` model, we plot the weights norm of each client's corresponding linear heads in the server in Figure 3 row 2. Combining it with row 1, we find that *the client with important local features indeed results in high weights.* For example, clients 6,7,8 in MNIST holding middle rows of images that contain the center of digits, have high weights, while clients 1, 14 holding the black background pixels have low weights. A similar phenomenon is observed on CIFAR for client 5 (center) and client 1 (corner). On ModelNet40, clients have complementary views of the same objects, so their features have similar importance, leading to similar weights norms. Based on our observation, we conclude that *the weights of linear heads can reflect the importance of local clients.* We use this principle to infer that, for NUS-WIDE, the first 500 dim. of textual features have higher importance than other multimodality features, resulting in the high weights norm of client 3.

**Client Importance Validation via Noisy Test Client.** Given a trained `VIMADMM` model, we add Gaussian noise to the test local features to verify the client-level importance indicated by the linear heads. For each time, we only perturb the features of one client and keep other clients' features unchanged. The results in Figure 3 row 3 show that *perturbing the client with high weights affects more for the test accuracy*, which verifies that clients with higher weights are more important.

**Client Denoising.** We study the denoising ability of `VIM` under training-time noisy clients. We construct one noisy client (i.e., client 7, 5, 2, 3 for MNIST, CIFAR, NUS-WIDE, ModelNet40 respectively) by adding Gaussian noise to its local features and re-train the `VIMADMM` model. The obtained weights norm in Figure 3 row 4 shows that `VIMADMM` can *automatically detect the noisy client and lower its weights* (compared to the clean one in Figure 3 row 2). Table 4 in Appendix B shows that under the noisy training scenario, `VIMADMM` and `VIMSGD` outperform VAFL with faster convergence and higher test accuracy.

## 5 Conclusions

In this work, we propose an efficient VFL framework with multiple heads (`VIM`). To solve our optimization problem, we propose an ADMM-based method for efficient communication. Extensive experiments verify the superior performance of our algorithms, and show that `VIM` enables client-level explainability.

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

# Contents

## A  Algorithm Details

### A.1  VIMSGD

At each communication round $t$, the server samples a set of data indices, $B(t)$, with batch size $|B(t)| = b$. Then we describe the key steps of VIMSGD as follows:

**(1) *Communication from client to server.*** Each client $k$ sends a batch of embeddings $\{h_j^{k(t)}\}_{j \in B(t)}$ to the server, where $h_j^{k(t)} = f(x_j^k; \theta_k^{(t)}), \forall j \in B(t)$.

**(2) *Sever updates linear heads* $\{W_k\}$.** According to VIM objective in Eq. 2, each linear head of the server is updated as:

$$W_k^{(t+1)} \leftarrow W_k^{(t)} - \eta \nabla_{W_k^{(t)}} \mathcal{L}_{\text{VIM}}(W_k^{(t)}), \forall k \in [M] \tag{11}$$

where $\eta$ is the server learning rate, and

$$\nabla_{W_k^{(t)}} \mathcal{L}_{\text{VIM}}(W_k^{(t)}) = \nabla_{W_k^{(t)}} \left( \sum_{j=1}^N \ell(\sum_{i=1}^M h_j^{i(t)} W_i^{(t)}, y_j) + \beta \mathcal{R}(W_k^{(t)}) \right) \tag{12}$$

**(3) *Communication from server to client.*** Server computes gradients w.r.t each local embedding $\nabla_{h_j^{k(t)}} \mathcal{L}_{\text{VIM}}(W_k^{(t+1)})$ by the VIM objective in Eq. 2, where

$$\nabla_{h_j^{k(t)}} \mathcal{L}_{\text{VIM}}(W_k^{(t+1)}) = \nabla_{h_j^{k(t)}} \ell(\sum_{i=1}^M h_j^{i(t)} W_i^{(t+1)}, y_j), \forall j \in B(t), k \in [M] \tag{13}$$

Server sends gradients $\{\nabla_{h_j^{k(t)}} \mathcal{L}_{\text{VIM}}(W_k^{(t+1)})\}_{j \in B(t)}$ to each client $k, \forall k \in [M]$.

**(4) *Client updates local model parameters* $\theta_k$.** Finally, every client $k$ locally updates the model parameters $\theta_k$ according to the VIM objective in Eq. 2 as follows:

$$\theta_k^{(t+1)} = \theta_k^{(t)} - \eta^k \nabla_{\theta_k^{(t)}} \mathcal{L}_{\text{VIM}}(W_k^{(t+1)}), \forall k \in [M] \tag{14}$$

where $\eta^k$ is the local learning rate for client $k$, and

$$\nabla_{\theta_k^{(t)}} \mathcal{L}_{\text{VIM}}(W_k^{(t+1)}) = \sum_{j=1}^N \nabla_{\theta_k^{(t)}} h_j^{k(t)} \nabla_{h_j^{k(t)}} \mathcal{L}_{\text{VIM}}(W_k^{(t+1)}) + \beta \nabla_{\theta_k^{(t)}} \mathcal{R}(\theta_k^{(t)}) \tag{15}$$

These four steps of VIMSGD are summarized in Algorithm 1.

### A.2  VIMADMM

We summarize the steps of VIMADMM are summarized in Algorithm 2.

### A.3  VIMADMM-J

At each communication round $t$, the server samples a set of data indices, $B(t)$, with batch size $|B(t)| = b$. Then we describe the key steps of VIMADMM-J as follows:

**(1) *Communication from client to server.*** Each client $k$ sends a batch of local logits $\{o_j^{k(t)}\}_{j \in B(t)}$ to the server, where $o_j^{k(t)} = f(x_j^k; \theta_k^{(t)}) W_k^{(t)}, \forall j \in B(t)$

**(2) *Sever updates auxiliary variables* $\{z_j\}$.** After receiving the local logits from all clients, the server updates the auxiliary variable for each sample $j$ as:

$$z_j^{(t)} = \underset{z_j}{\operatorname{argmin}} \quad \ell(z_j, y_j) - \lambda_j^{(t-1)^\top} z_j + \frac{\rho}{2} \left\| \sum_{k=1}^M o_j^{k(t)} - z_j \right\|_F^2, \forall j \in B(t) \tag{16}$$

---

**Algorithm 1:** `VIMSGD`

---

**Input:** number of communication rounds $T$, number of clients $M$, number of training samples $N$, batch size $b$ , input features $\{\{x_j^1\}_{j=1}^N, \{x_j^2\}_{j=1}^N, \ldots, \{x_j^M\}_{j=1}^N\}$, the labels $\{y_j\}_{j=1}^N$, local model $\{\theta_k\}_{k=1}^M$; linear heads $\{W_k\}_{k=1}^M$; server learning rate $\eta$; client learning rate $\{\eta^k\}_{k=1}^M$;

1 **for** *communication round* $t \in [T]$ **do**
2      Server samples a set of data indices $B(t)$ with $|B(t)| = b$
3      **for** *client* $k \in [M]$ **do**
4          **generates** a local training batch $\{x_j^k\}_{j \in B(t)}$
5          **computes** local embeddings $h_j^{k\,(t)} \leftarrow f(x_j^k; \theta_k), \forall j \in B(t)$
6          **sends** local embeddings $\{h_j^{k\,(t)}\}_{j \in B(t)}$ to the server
7      Server **updates** linear heads $W_k^{(t+1)}$ by Eq. 11 , $\forall k \in [M]$
8      Server **computes** gradients w.r.t embeddings $\nabla_{h_j^{k\,(t)}} \mathcal{L}_{\texttt{VIM}}(W_k^{(t+1)})$ by Eq. 13 , $\forall j \in B(t), k \in [M]$
9      Server **sends** gradients $\{\nabla_{h_j^{k\,(t)}} \mathcal{L}_{\texttt{VIM}}(W_k^{(t+1)})\}_{j \in B(t)}$ to each client $k, \forall k \in [M]$
10      **for** *client* $k \in [M]$ **do**
11          **updates** local model $\theta_k^{(t+1)}$ by Eq. 14

---

---

**Algorithm 2:** `VIMADMM`

---

**Input:** number of communication rounds $T$, number of clients $M$, number of training samples $N$, batch size $b$ , input features $\{\{x_j^1\}_{j=1}^N, \{x_j^2\}_{j=1}^N, \ldots, \{x_j^M\}_{j=1}^N\}$, the labels $\{y_j\}_{j=1}^N$, local model $\{\theta_k\}_{k=1}^M$; linear heads $\{W_k\}_{k=1}^M$; auxiliary variables $\{z_j\}_{j=1}^N$; dual variables $\{\lambda_j\}_{j=1}^N$;

1 **for** *communication round* $t \in [T]$ **do**
2      Server samples a set of data indices $B(t)$ with $|B(t)| = b$
3      **for** *client* $k \in [M]$ **do**
4          **generates** a local training batch $\{x_j^k\}_{j \in B(t)}$
5          **computes** local embeddings $h_j^{k\,(t)} \leftarrow f(x_j^k; \theta_k), \forall j \in B(t)$
6          **sends** local embeddings $\{h_j^{k\,(t)}\}_{j \in B(t)}$ to the server
7      Server **updates** auxiliary variables $z_j^{(t)}$ via Eq. 6 , $\forall j \in B(t)$
8      Server **updates** dual variables $\lambda_j^{(t)}$ via Eq. 7 , $\forall j \in B(t)$
9      Server **updates** linear heads $W_k^{(t+1)}$ with objective of Eq. 8 , $\forall k \in [M]$
10      Server **computes** residual variables $s_j^{k\,(t+1)}$ via Eq. 9 , $\forall j \in B(t), k \in [M]$
11      Server **sends** $\{\lambda_j^{(t)}\}_{j \in B(t)}$ , $\{s_j^{k\,(t+1)}\}_{j \in B(t)}$ and corresponding $W_k^{(t+1)}$ to each client $k, \forall k \in [M]$
12      **for** *client* $k \in [M]$ **do**
13          **for** *local step* $e \in [\tau]$ **do**
14              **updates** local model $\theta_k^{(t+1)}$ via SGD with objective of Eq. 10

---

Since the optimization problem in Eq. 16 is convex and differentiable with respect to $z_j$, we use the L-BFGS-B algorithm [37] to solve the minimization problem.

**(3)** *Sever updates dual variables* $\{\lambda_j\}$**.** After the updates in Eq. 16, the server updates the dual variable for each sample $j$ as:

$$\lambda_j^{(t)} = \lambda_j^{(t-1)} + \rho \left( \sum_{k=1}^M o_j^{k\,(t)} - z_j^{(t)} \right), \forall j \in B(t) \tag{17}$$

**(4)** *Communication from server to client.* After the updates in Eq. 17, we define a residual variable $s_j^{k\,(t+1)}$ for each sample $j$ of $k$-th client, which provides supervision for updating local model:

$$s_j^{k\,(t+1)} \triangleq z_j^{(t)} - \sum_{i \in [M], i \neq k} o_j^{i\,(t)} \tag{18}$$

**Algorithm 3:** `VIMADMM-J`

---

**Input:** number of communication rounds $T$, number of clients $M$, number of training samples $N$, batch size $b$, input features $\{\{x_j^1\}_{j=1}^N, \{x_j^2\}_{j=1}^N, \ldots, \{x_j^M\}_{j=1}^N\}$, the labels $\{y_j\}_{j=1}^N$, local model $\{\theta_k\}_{k=1}^M$; linear heads $\{W_k\}_{k=1}^M$; auxiliary variables $\{z_j\}_{j=1}^N$; dual variables $\{\lambda_j\}_{j=1}^N$;

1 **for** *communication round* $t \in [T]$ **do**
2      Server samples a set of data indices $B(t)$ with $|B(t)| = b_s$
3      **for** *client* $k \in [M]$ **do**
4          **generates** a local training batch $\{x_j^k\}_{j \in B(t)}$
5          **computes** local logits $o_j^{k^{(t)}} = f(x_j^k; \theta_k^{(t)}) W_k^{(t)}, \forall j \in B(t)$
6          **sends** local logits $\{o_j^{k^{(t)}}\}_{j \in B(t)}$ to the server
7      Server **updates** auxiliary variables $z_j^{(t)}$ via Eq. 16, $\forall j \in B(t)$
8      Server **updates** dual variables $\lambda_j^{(t)}$ via Eq. 17, $\forall j \in B(t)$
9      Server **computes** residual variables $s_j^{k^{(t+1)}}$ via Eq. 18, $\forall j \in B(t), k \in [M]$
10      Server **sends** $\{\lambda_j^{(t)}\}_{j \in B(t)}$, $\{s_j^{k^{(t+1)}}\}_{j \in B(t)}$ to each client $k, \forall k \in [M]$
11      **for** *client* $k \in [M]$ **do**
12          **for** *local step* $e \in [\tau]$ **do**
13              **updates** local linear head $W_k^{(t+1)}$ via SGD with objective of Eq. 19
14              **updates** local model $\theta_k^{(t+1)}$ via SGD with objective of Eq. 20

---

The server sends the dual variables $\{\lambda_j^{(t+1)}\}_{j \in B(t)}$ and the residual variables $\{s_j^{k^{(t+1)}}\}_{j \in B(t)}$ of all samples to each client $k$.

**(5)** *Client updates linear head $W_k$ and local model $\theta_k$ alternatively.* The linear head of each client is locally updated as:

$$W_k^{(t+1)} = \operatorname*{argmin}_{W_k} \quad \beta \mathcal{R}(W_k) + \sum_{j \in B(t)} \lambda_j^{(t)\top} f(x_{j_k}; \theta_k^{(t)}) W_k + \sum_{j \in B(t)} \frac{\rho}{2} \left\| s_j^{k^{(t+1)}} - f(x_{j_k}; \theta_k^{(t)}) W_k \right\|_F^2, \forall k \in [M]$$

$$(19)$$

Each client updates the local model parameters $\theta_k$ as follows:

$$\theta_k^{(t+1)} = \operatorname*{argmin}_{\theta_k} \quad \beta \mathcal{R}(\theta_k) + \sum_{j \in B(t)} \lambda_j^{(t)\top} f(x_{j_k}; \theta_k) W_k^{(t+1)} + \sum_{j \in B(t)} \frac{\rho}{2} \left\| s_j^{k^{(t+1)}} - f(x_{j_k}; \theta_k) W_k^{(t+1)} \right\|_F^2.$$

$$(20)$$

Due to the nonconvexity of the loss function of DNN, we use $\tau$ local steps of SGD to update $W_k$ and $\theta_k$ alternatively at each round with the objective of Eq. 19 and Eq. 20. Specifically, at each local step, we first update $W_k$ and then update $\theta_k$.

These five steps of `VIMADMM-J` are summarized in Algorithm 3.

# B   Experimental Details

## B.1   Datasets and Models

We consider a diverse set of datasets and tasks.

- MNIST [22] contains images with handwritten digits. We create the VFL scenario by splitting the input features evenly by rows for 14 clients. We use a fully connected model of two linear layers with ReLU activations as the local model.

- CIFAR [21] contains colour images. We split each image into patches for 9 clients. We use a standard CNN architecture from the PyTorch library [1] as the local model.

- NUS-WIDE [6] is a multi-modality dataset with 634 low-level image features and 1000 textual tag features. We distribute image features to 2 clients (300 dim and 334 dim), and

---

[1]https://github.com/pytorch/opacus

text features to 2 clients (500 dim and 500 dim). We use a fully connected model of two linear layers with ReLU activations as the local model.

- ModelNet40 [27] is a multi-view image dataset, containing the shaded images from 12 views for the same objects. We use 4 views and distribute them to 4 clients respectively. We use ResNet-18 [14] as the local model.

We split each dataset into the train, validation, and test sets. See Table 1 for more details about the number of samples and the number of classes for each dataset.

## B.2 Platform

We simulate the vertical federated learning setup (1 server and N users) on a Linux machine with AMD Ryzen Threadripper 3990X 64-Core CPUs and 4 NVIDIA GeForce RTX 3090 GPUs. The algorithms are implemented by PyTorch [25]. Please see the submitted code for full details. We run each experiment 3 times with different random seeds.

## B.3 Hyperparameters

We detail our hyperparameter tuning protocol and the hyperparameter values here. For all VFL training experiments, we use the SGD optimizer with learning rate $\eta$ for the server's model, and the SGD optimizer with momentum 0.9 and learning rate $\eta^k$ for client $k$'s local model. We set $\eta = \eta^1, \eta^2, \ldots, \eta^M$ for all methods. The regularization weight $\beta$ is set to 0.005. The embedding dimension $d_f$ is set to 60, and batch size $b$ is set to 1024 for all datasets.

**Vanilla VFL Training** For Vanilla VFL training experiments, we tune learning rates by performing a grid search separately for all methods over $\{0.1, 0.3, 0.5, 0.8\}$ on MNIST, $\{0.003, 0.005, 0.008, 0.01, 0.05, 0.1\}$ on CIFAR, $\{0.1, 0.5\}$ on NUS-WIDE, $\{0.0005, 0.005, 0.01, 0.05, 0.1\}$ on ModelNet40. Table 1 summarize hyperparameters for all methods.

Table 1: Dataset description and hyperparameters for Vanilla VFL Training.

| Dataset | # features | $d_c$ | $M$ | # samples train | # samples validation | test | VAFL $\eta$ | VIMSGD $\eta$ | VIMADMM $\eta$ | VIMADMM $\rho$ | VIMADMM $\tau$ | FDML $\eta$ | VIMADMM-J $\eta$ | VIMADMM-J $\rho$ | VIMADMM-J $\tau$ |
|---|---|---|---|---|---|---|---|---|---|---|---|---|---|---|---|
| MNIST | $28 \times 28$ | 10 | 14 | 54000 | 6000 | 10000 | 0.3 | 0.3 | 0.05 | 2 | 20 | 0.1 | 0.05 | 0.5 | 20 |
| CIFAR | $32 \times 32 \times 3$ | 10 | 9 | 45000 | 5000 | 10000 | 0.003 | 0.005 | 0.005 | 2 | 30 | 0.005 | 0.005 | 2 | 30 |
| NUS-WIDE | 1634 | 5 | 4 | 54000 | 6000 | 10000 | 0.1 | 0.5 | 0.05 | 2 | 20 | 0.1 | 0.05 | 2 | 20 |
| ModelNet40 | $224 \times 224 \times 3 \times N$ | 40 | 4 | 8877 | 966 | 2468 | 0.05 | 0.05 | 0.05 | 0.5 | 5 | 0.05 | 0.05 | 0.5 | 5 |

**Client-level Explainability** In the experiments of *client importance validation via noisy test client*, for each time, we perturb the features of all test samples at one client by adding Gaussian noise sampled from $\mathcal{N}\left(0, \bar{\sigma}^2\right)$ to its features. In order to observe the difference in test accuracy between important clients and unimportant clients, we set $\bar{\sigma}$ to 10 for MNIST, 1 for CIFAR and NUS-WIDE, and 3 for ModelNet40.

In the experiments of *client denoising*, we construct one noisy client (i.e., client 7, 5, 2, 3 for MNIST, CIFAR, NUS-WIDE, ModelNet40 respectively) by adding Gaussian noise sampled from $\mathcal{N}\left(0, \widetilde{\sigma}^2\right)$ to all its training samples and test samples. We set $\widetilde{\sigma}$ to 1 for MNIST, NUS-WIDE and ModelNet40, and 3 for CIFAR.

## B.4 Additional Results

**Comparison under Communication Cost.** Here we report the memory of parameters communicated between clients and the server to evaluate communication cost. We use batch size 1024 and local embedding size 60 for all datasets following the hyper-parameters listed in Table 1.

Table 2 shows that for each round, VAFL, VIMSGD and VIMADMM have the same number of parameters sent from each client to the server (i.e., 0.23 MB for a batch of embeddings), and VIMADMM has a smaller number of parameters sent from server to each client (i.e., 0.08 MB in total for a batch of

dual variables, residual variables as well as one corresponding linear head) than VAFL and `VIMSGD` (i.e., 0.23 MB for a batch of gradients w.r.t. embeddings).

Table 2 and Figure 4 also show that `VIMADMM` requires significantly lower communication costs to reach a target performance. For example, in CIFAR, to achieve a target accuracy of 65.0%, VAFL needs 9463.85 MB while `VIMADMM` only requires 124.54 MB, which is about 76x lower costs.

| Method | Communication costs (MB) per round | | | Communication costs (MB) to reach target performance | | | |
|---|---|---|---|---|---|---|---|
| | Each client to server | Server to each client | Total | MNIST ($\geq 96.5\%$) | CIFAR ($\geq 65.0\%$) | NUS-WIDE ($\geq 85.0\%$) | ModelNet40 ($\geq 89.0\%$) |
| VAFL | 0.23 | 0.23 | 0.46 | 6954.02 | 9463.85 | 695.40 | 134.96 |
| VIMSGD | 0.23 | 0.23 | 0.46 | 3824.71 | 5381.40 | 198.69 | 84.35 |
| VIMADMM | 0.23 | **0.08** | **0.31** | **700.08** | **124.54** | **66.67** | **11.32** |

Table 2: Communication cost comparison.

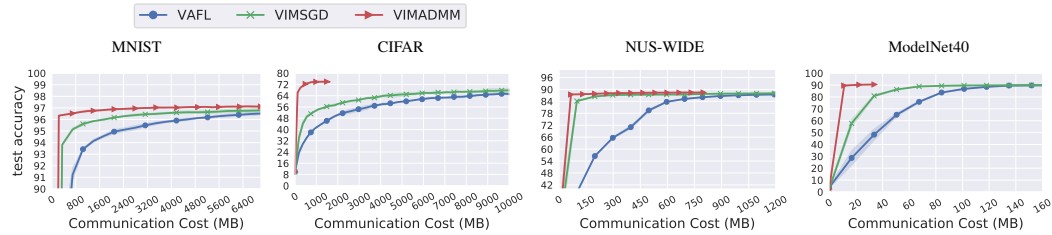

Figure 4: Performance of Vanilla VFL under w/ model splitting setting. Ours consume significantly lower communication costs to reach a target performance.

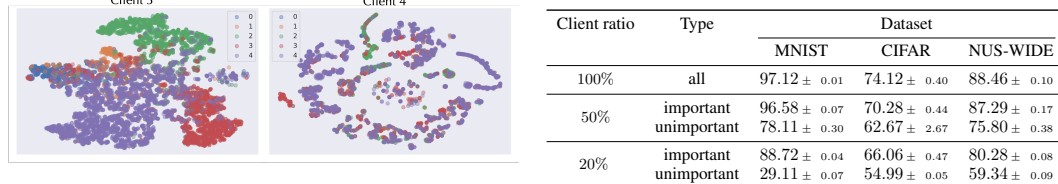

| Client ratio | Type | Dataset | | |
|---|---|---|---|---|
| | | MNIST | CIFAR | NUS-WIDE |
| 100% | all | $97.12_{\pm\ 0.01}$ | $74.12_{\pm\ 0.40}$ | $88.46_{\pm\ 0.10}$ |
| 50% | important | $96.58_{\pm\ 0.07}$ | $70.28_{\pm\ 0.44}$ | $87.29_{\pm\ 0.17}$ |
| | unimportant | $78.11_{\pm\ 0.30}$ | $62.67_{\pm\ 2.67}$ | $75.80_{\pm\ 0.38}$ |
| 20% | important | $88.72_{\pm\ 0.04}$ | $66.06_{\pm\ 0.47}$ | $80.28_{\pm\ 0.08}$ |
| | unimportant | $29.11_{\pm\ 0.07}$ | $54.99_{\pm\ 0.05}$ | $59.34_{\pm\ 0.09}$ |

Figure 5: T-SNE of embeddings on NUS-WIDE.    Table 3: Client summarization of `VIMADMM`.

**T-SNE of Local Embeddings.** From the T-SNE [28] visualizations in Figure 5, we show that client 3 produces linear separable local embeddings (left), which are better than client 4's embeddings (right) that overlap different classes. Therefore, the embedding averaging from VAFL [4] is suboptimal, which justifies the design of `VIM`, taking the properties of different local embeddings into account.

Figure 6 presents the T-SNE visualizations of local embeddings for the model trained from `VIMADMM`. Similar to the results of NUS-WIDE in Figure 5, Figure 6 shows that important clients learn better local embeddings than unimportant clients on MNIST and CIFAR, which justifies our design of multiple linear heads in `VIM`. For ModelNet40, since clients with multi-view data are of similar importance, their local embeddings are similar and are linearly separable.

**Client Summarization.** We study the functionality of client summarization enabled by `VIM`. (1) We first rank the importance of clients according to the weights norm histogram (i.e., Figure 3 row 2), then we select $u\%$ proportion of the most "important" clients to re-train the `VIMADMM` model. We find that *its performance is closed to the one trained by all clients.* Table 3 shows that the test accuracy-drop of training with 50% of the most important clients is less than 1% on MNIST and NUS-WIDE, and less than 4% on CIFAR; the accuracy-drop of training with 20% of the most important clients is less than 10% on all datasets. (2) We select $u\%$ proportion of the least important clients to re-train the model, and we find that its performance is significantly lower than the one trained with important clients, which indicates the effectiveness of `VIM` for client selection. (3) For the multi-view dataset ModelNet40, we find that the test accuracy of models trained with 12, 8, and 4 clients are similar, i.e., 91.04%, 90.69%, and 90.64%, suggesting that a few views can already provide sufficient training information and the agents with multiview data are of similar importance which is also reflected by our linear head weights.

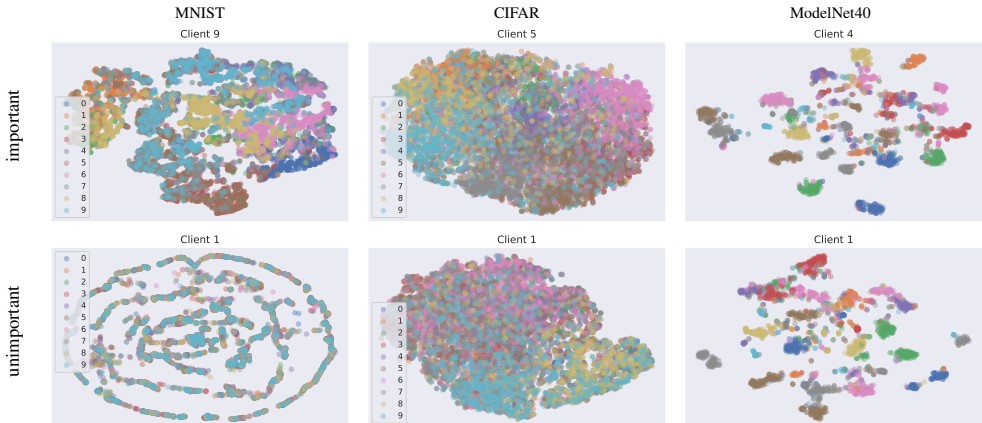

Figure 6: T-SNE visualizations of local embeddings from important client and unimportant client for `VIMADMM`.

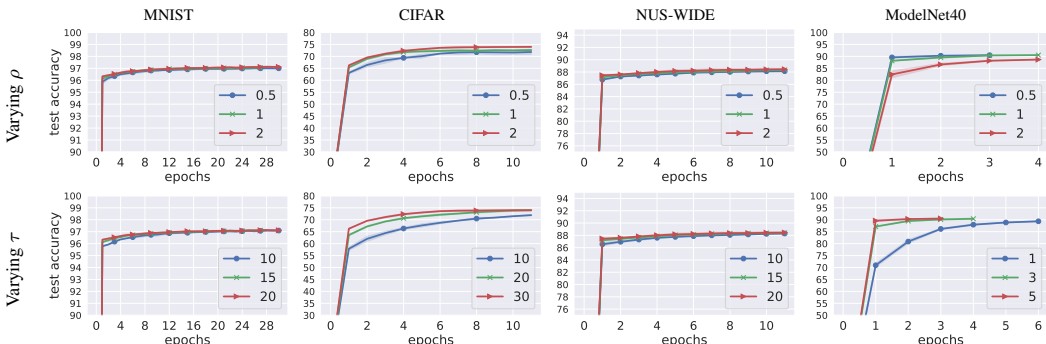

Figure 7: Performance of `VIMADMM` with different penalty factor $\rho$ on four datasets.

**Effect of Penalty Factor $\rho$ and Local Steps $\tau$ for `VIMADMM`** The results in Figure 7 first row show that `VIMADMM` is not sensitive to $\rho$ on datasets, and we suggest that the practitioners choose the optimal $\rho$ from 0.5 to 2, which will not influence the test accuracy significantly. The results in Figure 7 second row show that when $\tau$ is larger, the `VIMADMM` algorithm converges faster. This is because the local models can be trained better with more local update steps (i.e., larger $\tau$) at each communication round. Therefore, we suggest that the practitioners choosex a $\tau$ that leads to the converged local model at each communication round.

**Additional Results on Client Denoising** Table 4 presents the test accuracy of VAFL, `VIMSGD`, and `VIMADMM` at different epochs (communication rounds) on different datasets under one noisy client. Note that each epoch consists of $N/b$ communication rounds. Table 4 shows that under the noisy training scenario, `VIMADMM` and `VIMSGD` consistently outperform VAFL with faster convergence and higher test accuracy, which indicates the effectiveness of `VIM`'s multiple linear heads in client denoising.

Table 4: Test accuracy under one noisy client whose training local features and test local features are perturbed by Gaussian noise.

| Method | Test accuracy @ epoch (communication round) | | | | | | | | | | | |
|---|---|---|---|---|---|---|---|---|---|---|---|---|
| | MNIST | | | CIFAR | | | NUS-WIDE | | | ModelNet40 | | |
| | 2 (106) | 5 (265) | 10 (530) | 2 (88) | 5 (220) | 10 (440) | 2 (106) | 5 (265) | 10 (530) | 2 (18) | 5 (45) | 10 (90) |
| VAFL | $91.07 \pm 0.17$ | $94.36 \pm 0.16$ | $95.59 \pm 0.11$ | $28.83 \pm 1.04$ | $38.77 \pm 0.39$ | $46.98 \pm 0.70$ | $51.88 \pm 0.72$ | $77.68 \pm 0.74$ | $85.31 \pm 0.15$ | $43.23 \pm 3.07$ | $80.13 \pm 1.10$ | $89.56 \pm 0.41$ |
| VIMSGD | $95.04 \pm 0.14$ | $96.01 \pm 0.03$ | $96.43 \pm 0.08$ | $42.75 \pm 0.13$ | $50.06 \pm 0.18$ | $55.53 \pm 0.37$ | $85.35 \pm 0.24$ | $86.42 \pm 0.24$ | $87.14 \pm 0.29$ | $77.94 \pm 1.00$ | $88.74 \pm 0.07$ | $89.69 \pm 0.42$ |
| VIMADMM | $96.22 \pm 0.07$ | $96.60 \pm 0.04$ | $96.82 \pm 0.07$ | $67.08 \pm 0.43$ | $70.70 \pm 0.34$ | $71.76 \pm 0.14$ | $86.38 \pm 0.20$ | $87.00 \pm 0.27$ | $87.18 \pm 0.14$ | $90.05 \pm 0.38$ | $90.71 \pm 0.31$ | $90.59 \pm 0.05$ |

**More results for a large number of clients.** We evaluate baselines and our methods under 100 clients on MNIST by allowing the agents to obtain overlapped features, and the results show that our

methods still outperform baselines. Specifically, we divide the features into 100 overlapped subsets for 100 clients so that each client has 14 pixels. We train the methods using the hyper-parameters setup listed in Table 1.

The results in Table 5 show that VIM methods (i.e., `VIMSGD`, `VIMADMM`, `VIMADMM-J`) have higher accuracy than baselines in both w/ and w/o model splitting settings.

| W/ model splitting | | | W/o model splitting | |
|---|---|---|---|---|
| VAFL | VIMSGD | VIMADMM | FDML | VIMADMM-J |
| 95.38 | 95.45 | **95.77** | 95.85 | **95.96** |

Table 5: Performance of Vanilla VFL when $M = 100$ on MNIST

## C   Discussion

**Challenges of ADMM in VFL.**   There are several key challenges of deploying ADMM in VFL for distributed optimization:

(1) how to ensure the consensus among clients and form it as a constrained optimization problem (e.g., from Eq. 2 to Eq. 3);

(2) how to decompose the optimization problem into small sub-problems that can be solved in parallel by ADMM (e.g., from Eq. 3 to Eq. 5).

For the first challenge, although ADMM is flexible to introduce auxiliary variables and thus formulate a constrained optimization problem in HFL, it raises new challenges in VFL. For example, the ADMM-based methods in HFL [8, 7, 17, 35] usually use the global model as the auxiliary variable and enforce the consistency between the global model and each local model. However, VFL communicates embeddings, and it is not feasible to enforce local embeddings from different clients to be the same as they provide unique information from different aspects. Therefore, in this paper, we introduce the auxiliary variable $z_j$ for each sample $j$ and construct the constraint between $z_j$ and server's output $\sum_{k=1}^{M} h_j^k W_k$ (i.e., the logits), which enables the optimization for each $W_k$ by ADMM (i.e., Eq. 5).

For the second challenge, we propose the bi-level optimization for server's model and clients' models to train DNNs for VFL with model splitting, while the existing ADMM-based method in VFL [15] only considers logistic regression with linear models in client-side, which does not apply to DNNs. The initial attempt we made is to decompose the optimization for server's linear heads by ADMM while still using chain rule of SGD to update local models, which does not exhibit much superiority over pure SGD-based methods. Later, we decompose the optimization for both server's linear heads and local models by ADMM, leading to our current algorithm `VIMADMM` that enables multiple local updates for clients at each communication round and achieves significantly better performance as we show in Sec. 4.1.

