# OpenReview forum: "Improving Vertical Federated Learning by Efficient Communication with ADMM"
_NeurIPS.cc/2022/Workshop/Federated_Learning — FL-NeurIPS 2022 Poster_

### Official Review · Reviewer_fGnT · 2022-10-17
**Lack novelty, experimental details and important comparison**

This paper considers the multiple linear heads on the server in vertical federated learning and evaluates this design on several datasets.

1. The idea is very straightforward and lacks novelty. Since in vertical federated learning, partial feature sets are stored in different clients, it's obvious to consider two different designs: a) average learned features; b) expanding the feature-length on the server side. I didn't check enough related works in this area, but the authors claim only the former was considered and use option (b). I'm not convinced this design is novel enough to be accepted at this moment.

2. Lack of experimental details and lack of comparison. Even though the Appendix. B is the experimental details; crucial designs are still missing, such as how the networks are designed for different clients. Do you consider some ablations with different client networks? The authors mentioned they use local training and aggregation; how does this design generally influence the final performance? How do different local training methods impact the design (b) mentioned in point 1? More experiments should also be considered to evaluate the more general advantage of the proposed design.

---

### Official Review · Reviewer_DMtb · 2022-10-19
**good paper. some of the mathematical symbols are not well defined though**

this is a good paper, but some of the mathematical notations are not defined too well in the paper.

---

### Official Review · Reviewer_sH9S · 2022-10-19

This paper proposed a new method to overcome two issues in previous vertical federated learning algorithms. In particular, previous methods simply averages embeddings from different clients. However, these embeddings may not share the same latent space. In order to address this issue, the author proposed to enable different linear head for different clients at the server, and only averages the model predictions. Moreover, previous VFL algorithms requires gradient communication at each iteration. The authors reduced the communication cost by enabling local updates through an ADMM-based framework. Experiments on four datasets validated the effectiveness of the proposed method.

Overall, I believe this paper provides some new insights, has solid experiments, and worth presenting at the workshop.

=====Pros=====
- The proposed method addresses two critical issues in VFL.
- As a side benefit, the proposed method can automatically outputs client importance.


=====Cons=====
- Lack of theoretical guarantee.

---

### Decision · Program_Chairs · 2022-10-20

Accept (Poster)